# SEMANTIC INTERPOLATION IN IMPLICIT MODELS

**Yannic Kilcher, Aurélien Lucchi, Thomas Hofmann**
Department of Computer Science
ETH Zurich
{yannic.kilcher,aurelien.lucchi,thomas.hofmann}@inf.ethz.ch

## ABSTRACT

In implicit models, one often interpolates between sampled points in latent space. As we show in this paper, care needs to be taken to match-up the distributional assumptions on code vectors with the geometry of the interpolating paths. Otherwise, typical assumptions about the quality and semantics of in-between points may not be justified. Based on our analysis we propose to modify the prior code distribution to put significantly more probability mass closer to the origin. As a result, linear interpolation paths are not only shortest paths, but they are also guaranteed to pass through high-density regions, irrespective of the dimensionality of the latent space. Experiments on standard benchmark image datasets demonstrate clear visual improvements in the quality of the generated samples and exhibit more meaningful interpolation paths.

## 1 INTRODUCTION

Continuous latent variable models have been developed and studied in statistics for almost a century, with factor analysis (Young (1941); Bartholomew (1987)) being the most paradigmatic and widespread model family. In the neural network community, autoencoders have been used to find low-dimensional codes with low reconstruction error (Baldi & Hornik (1989); DeMers & Cottrell (1993)). Recently there has been an increased interest in *implict models*, where a complex generative mechanism is driven by a source of randomness (cf. MacKay (1995)). This includes popular architectures known as Variational Autoencoders (VAEs, Kingma & Welling (2013); Rezende et al. (2014)) as well as Generative Adversarial Networks (GANs, Goodfellow et al. (2014)). Typically, one defines a deterministic mechanism or generator $G_\phi : \mathbb{R}^d \to \mathbb{R}^m$, $\mathbf{z} \mapsto G_\phi(\mathbf{z}) = \mathbf{x}$, parametrized by $\phi$ and often implemented as a deep neural network (DNN). This map is then hooked up to a code distribution $\mathbf{z} \sim \mathcal{P}_\mathbf{z}$, to induce a distribution $\mathbf{x} \sim \mathcal{P}_\mathbf{x}$. It is known that under mild regularity conditions, by a suitable choice of generator, any $\mathcal{P}_\mathbf{x}$ can be obtained from an arbitrary *fixed* $\mathcal{P}_\mathbf{z}$ (cf. Kallenberg (2006)). Relying on the representational power and flexibility of DNNs, this has led to the view that code distributions should be simple, e.g. most commonly $\mathcal{P}_\mathbf{z} = \mathcal{N}(\mathbf{0}, \mathbf{I})$. Implicit models are essentially sampling devices that can be trained and used without explicit access to the densities they define. They have shown great promise in producing samples that are perceptually indistinguishable from samples generated by nature (Radford et al. (2015)) and are currently a subject of extensive investigation.

Earlier work on embedding models such as the word embeddings of Mikolov et al. (2013a), has shown how semantic relations and analogies are naturally captured by the affine structure of embeddings (Mikolov et al. (2013c); Levy & Goldberg (2014)). This has inspired the use of affine vector arithmetic and linear interpolation in implicit models such as GANs, where it has shown to lead to semantic interpolations in image space (cf. Radford et al. (2015); White (2016)). These *traversal* experiments have also been used to justify that deep generative models do not only memorize the training data, but do learn models that generalize to unseen data. However, as pointed out by White (2016), the commonly used linear interpolation has one major flaw in that it ignores the manifold structure of the latent space. Indeed, traversing the latent space along straight lines may lead through low-density regions, where – by definition – the generator has not been trained (well). This is easy to understand as for $\mathbf{z} \sim \mathcal{N}(\mathbf{0}, \mathbf{I})$ we get that

$$\|\mathbf{z}\|^2 \sim \chi^2(d), \quad \text{and hence} \quad \mathbf{E}\|\mathbf{z}\|^2 = d, \quad \text{Var}\left[\frac{\|\mathbf{z}\|^2}{d}\right] = \frac{2}{d}. \tag{1}$$

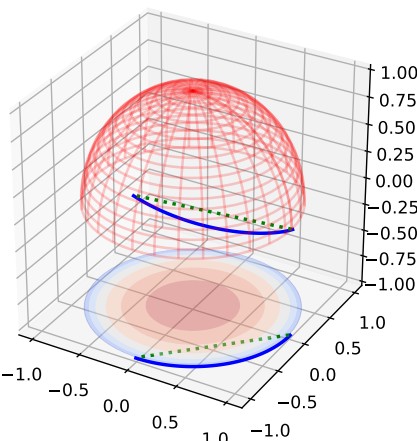

Figure 1: Illustration showing the distribution of the mass for a Normal prior distribution. In high-dimensions, most the mass concentrates in the outer shell shown in blue while the middle shown in red is almost massless. Linearly interpolating between 2 points produces the green dotted trajectory thus traversing a region of the space with low probability mass. In contrast, following the great circle arc in blue produces samples that are more likely but these paths can also be unstable as they exhibit higher variance (see main text).

Relative to the average, the variance of the squared code vector norm vanishes like $\mathbf{O}(\frac{1}{d})$ with the latent space dimensionality. Thus as $d$ increases, the probability mass concentrates in a thin shell around the $(d-1)$-sphere with radius $\sqrt{d}$. As shown in Figure 1, the interpolation paths between any two points on this sphere will always pass through the interior, and the larger their distance, the closer the path's midpoint will be to the origin.

Although we are not the first to point out this weakness, there has not been any rigorous attempts to analyze this phenomenon and to come up with a substantially improved interpolation scheme. The proposal by White (2016) is to use spherical linear interpolation. However, note that this scheme produces interpolation curves that are very similar to great circle arcs. These paths can be unstable (think of slightly perturbing points at opposing poles of the hyper-sphere) as well as unnecessarily long, passing through codes of images, say, that have very little in common with the ones at either endpoint.

Towards this goal, we make the following contributions:

i) We properly analyze the phenomenon by characterizing the KL-divergence between the latent code prior and the effective distribution induced on interpolated in-between points.

ii) We propose a modified prior distribution that effectively puts more probability mass closer to the origin and that provably controls the KL-divergence in a dimension-independent manner.

iii) We argue that linear interpolation by straight lines in this new model does not suffer from the problem identified in the original model.

iv) We provide extensive experiments that demonstrate the different nature of the interpolation paths and that show clear evidence of improved visual quality of in-between samples for images.

## 2 METHOD

### 2.1 NAIVE SAMPLING FROM A NORMAL DISTRIBUTION

**Distributional mismatch** We consider the common GAN framework with generator $G$ and latent code vectors sampled from an isotropic normal distribution, i.e. $\mathbf{z} \sim \mathcal{N}(\mathbf{0}, \sigma^2 \mathbf{I})$. In the typical traversal experiment, one considers two code vectors $\mathbf{z}_0, \mathbf{z}_1 \in \mathbb{R}^d$ sampled independently and interpolates between them with a straight line $h : [0;1] \rightarrow \mathbb{R}^d$, $t \mapsto h(t) = (1-t)\mathbf{z}_0 + t\mathbf{z}_1$. In doing so, one expects, for instance, that the mid-point $\mathbf{z}' = h(\frac{1}{2})$ should correspond to a sample that semantically relates to both $G(\mathbf{z}_0)$ and $G(\mathbf{z}_1)$. However, based on the arguments made before, it is often found in practice that the code $h(\frac{1}{2})$ falls in a latent space region of low probability mass. Consequently, the generated samples are often not representative of the data distribution. To elucidate this further, note that the distribution of the squared norm of midpoints can be shown to follow a Gamma distribution, namely,

$$\|\mathbf{z}'\|^2 = \left\| \frac{\mathbf{z}_0 + \mathbf{z}_1}{2} \right\|^2 = \frac{2\sigma^2}{4} \|\mathbf{n}\|^2 \quad \text{where } \mathbf{n} \sim \mathcal{N}(\mathbf{0}, \mathbf{I}) \tag{2}$$

Thus,

$$\|\mathbf{z}'\|^2 \sim \frac{\sigma^2}{2} \chi^2(d) = \Gamma(\tfrac{d}{2}, \sigma^2) \tag{3}$$

In particular this implies that in expectation the norm of the midpoint is a factor of $\frac{1}{\sqrt{2}}$ smaller than the norm at the endpoints. What conclusion can we draw from this observation? Mainly that the process used to train the generator network and the evaluation strategy used when traversing the latent space are not consistent. Indeed, at training time, the generator network is trained with vectors whose squared norms follow a $\sigma^2 \chi^2(d)$ distribution. At test time, however, the traversal procedure passes through midpoints whose squared norms follow a $\frac{\sigma^2}{2} \chi^2(d)$ distribution. Clearly this leads to a problematic train-test mismatch.

**Formal Analysis** Before we formalize the observations we made so far, let us collect some properties of the $\chi^2$-distribution that are needed for the analysis below. In statistics, the $\chi^2$ distribution is usually introduced via Eq. (1). Its density has support on the non-negative reals and can shown to be given by

$$f_{\chi^2}(u; d) = \frac{1}{2^{\frac{d}{2}} \Gamma\left(\frac{d}{2}\right)} u^{\frac{d}{2}-1} e^{-\frac{u}{2}}, \quad \text{for } u \geq 0 \tag{4}$$

The $\sigma^2 \chi^2(d)$ distribution is a special case of a Gamma distribution $\Gamma\left(\frac{d}{2}, 2\sigma^2\right)$ with shape parameter $\frac{d}{2}$ and scale $2\sigma^2$. This generalization is helpful as we can now also identify the midpoint distribution as a Gamma distribution, namely $\Gamma(\frac{d}{2}, \sigma^2)$. The following lemma gives a characterization of the KL-divergence between these two latent space distributions.

**Lemma 1.** *Let* $\mathbf{z} \sim \Gamma(\frac{d}{2}, 2\sigma^2)$ *and* $\mathbf{z}' \sim \Gamma(\frac{d}{2}, \sigma^2)$ *for any* $\sigma > 0$*, then*

$$KL(\mathbf{z}\|\mathbf{z}') = \frac{d}{2}(1 - \log 2) \tag{5}$$

*Proof.* Using straightforward calculus. □

In summary, KL$(\mathbf{z}\|\mathbf{z}')$ strictly increases with $d$, growing linearly in $d$ with the given rate. However, as pointed out by Arjovsky & Bottou (2017), we need a sufficiently high latent space dimension that at least matches the intrinsic dimensionality of the data manifold. Otherwise it is impossible for $\mathcal{P}_{\mathbf{z}}$ to be continuous and then stability issues commonly observed with GANs may occur. Thus it seems hard to avoid the blow-up of the KL-divergence that comes with large $d$.

**Spherical interpolation**    One remedy to counteract the above problem is to use spherical interpolation, essentially generalizing the notion of geodesics on a hyper-sphere. We have already eluded to the proposal of White (2016), which uses the interpolation with $\theta := \angle(\mathbf{z}_0, \mathbf{z}_1)$,

$$h(t) = \frac{\sin((1-t)\theta)}{\sin(\theta)}\mathbf{z}_0 + \frac{\sin(t\theta)}{\sin(\theta)}\mathbf{z}_1, \quad \mathbf{z}' = h(\tfrac{1}{2}) = \frac{\sin(\frac{\theta}{2})}{\sin(\theta)}(\mathbf{z}_0 + \mathbf{z}_1) = \frac{1}{\cos(\frac{\theta}{2})}\frac{\mathbf{z}_0 + \mathbf{z}_1}{2}. \quad (6)$$

It is easy to check that if $\|\mathbf{z}_0\| = \|\mathbf{z}_1\| = r$, then this curve follows the great circle with radius $r$ that connects $\mathbf{z}_0$ and $\mathbf{z}_1$. In the more general case as long as $|\theta| \leq \pi/2$, we get the bounds $\min\{\|\mathbf{z}_0\|, \|\mathbf{z}_1\|\} \leq \|h(t)\| \leq \max\{\|\mathbf{z}_0\|, \|\mathbf{z}_1\|\}$. While this interpolation formula may be appropriate in the original context of animating rotations as in Shoemake (1985), it is known that for larger angles it does not lead to semantically meaningful interpolation paths for GANs as these paths get too long, often passing through images that are seemingly unrelated. In addition, spherical interpolation destroys the simple affine vector arithmetic that has proven so useful in other contexts and that has shown to disentangle the nonlinear factors of variation into simple linear statistics (Mikolov et al. (2013a)). Therefore, it has been our goal to fix the divergence problem in a way that allows us to stick to linear interpolation in code space.

## 2.2   GAMMA DISTANCE MODEL

There is nothing sacred about the isotropic normal distribution as a code vector distribution other than a certain non-informativeness in the absence of other requirements. Here we suggest to keep the isotropic nature of the latent space distribution, but to modify the distribution of the norm or distance from the origin. Thus we factor the distribution as follows:

$$\mathbf{v} \sim \text{Uniform}(\mathcal{S}^{d-1}), \quad r \sim \mathcal{P}_r, \quad \mathbf{z} = \sqrt{r}\,\mathbf{v}. \quad (7)$$

The choice $\mathcal{P}_r = \chi^2(d)$ brings us back to the normal case and the problems that come with it. Instead, we eliminate the dimension dependency in the choice of $\mathcal{P}_r$. One simple way to accomplish that is to stay within the family of $\Gamma$-distributions with fixed shape and set

$$\mathcal{P}_r = \Gamma(\tfrac{1}{2}, \theta), \quad \theta > 0. \quad (8)$$

In particular, if $\theta = 2$, this results in the same marginal distribution over norms than in the 1-dimensional Gaussian case, thereby counteracting the concentration effect on the hyper-sphere of radius $\sqrt{d}$.

**Proposition 1.** *Using the model described in Eqs.* (7)-(8)*, we have*

$$\|\mathbf{z}_0\|^2, \|\mathbf{z}_1\|^2 \sim \Gamma(\tfrac{1}{2}, \theta), \quad \left\|\frac{\mathbf{z}_0 + \mathbf{z}_1}{2}\right\|^2 \sim \Gamma(\tfrac{1}{2}, \tfrac{1}{2}\theta). \quad (9)$$

*Furthermore the KL divergence between $\mathbf{z}$ and mid points $\mathbf{z}'$ is given by*

$$KL(\mathbf{z}\|\mathbf{z}') = \frac{1}{2}(1 - \log 2) \approx 0.35. \quad (10)$$

What have we gained? We would like to make two observations: (i) Eq (10) shows that the KL divergence is constant and does not grow with the latent space dimension. (ii) While retaining a constant divergence, the gamma sampling procedure still offers the ability to tune the noise level through the free scale parameter $\theta$ (similar to $\sigma$ for the original sampling procedure).

## 3 EXPERIMENTS

**Experimental results.** The setup used for the experiments presented below closely follows popular setups in GAN research and is detailed in the Appendix.

### 3.1 SAMPLES FROM GAN WITH Γ-PRIOR

Figure 3.1 compares the samples generated from a Normal prior to the gamma prior for different benchmark image datasets. In addition to straightforwardly replacing the noise sampling, we used no additional tricks to obtain these results. This shows that GANs with gamma priors can be trained just as easily as traditional GANs.

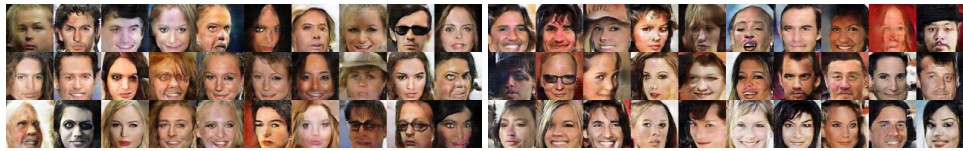

(a) Samples from CelebA with normal (left) and gamma (right) prior

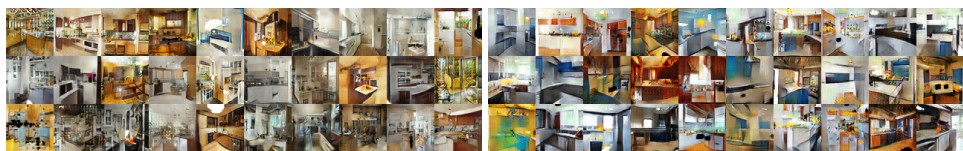

(b) Samples from LSUN kitchen with normal (left) and gamma (right) prior

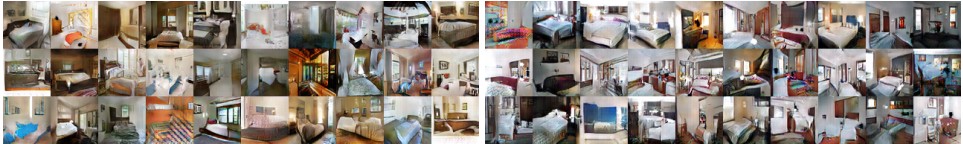

(c) Samples from LSUN bedroom with normal (left) and gamma (right) prior.

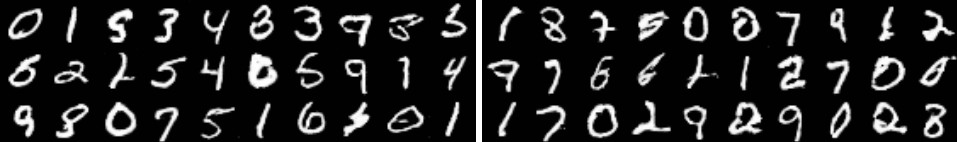

(d) Samples from MNIST with normal (left) and gamma (right) prior.

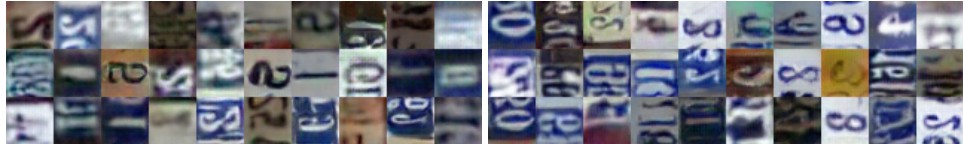

(e) Samples from SVHN with normal (left) and gamma (right) prior.

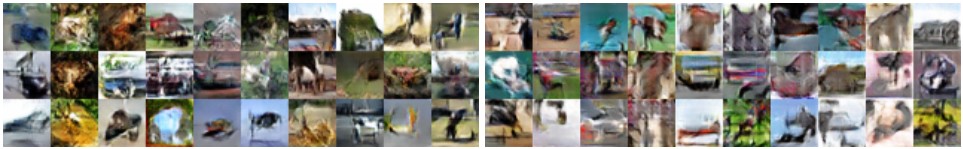

(f) Samples from CIFAR10 with normal (left) and gamma (right) prior.

## 3.2 TRAVERSAL EXPERIMENTS

Here we perform two different types of traversals from the same pair of points of same length lying on opposite sides of the center of the sphere. While one traversal goes straight (in a Euclidean sense) through the middle, the other traversal goes along a geodesic on the sphere.

We compare the two traversal paths for a model trained using a multivariate normal prior and for a model using our suggested gamma prior. Along with sampled traversal trajectories, we also show the discriminator activation along these trajectories, averaged over 1000 trajectory samples. Plotted are the mean discriminator activation and one standard deviation.

More traversal samples can be found in the Appendix.

### 3.2.1 SPHERE GEODESIC TRAVERSAL

Figures 2 and 3 show traversals and discriminator activation along a great circle on the sphere in latent space. Note that often, the path taken is visiting realistic and interesting samples, but is not semantically interpolating between the given pair of endpoints. Also note that the discriminator activation stays the same along the paths, meaning it judges all samples as equally likely.

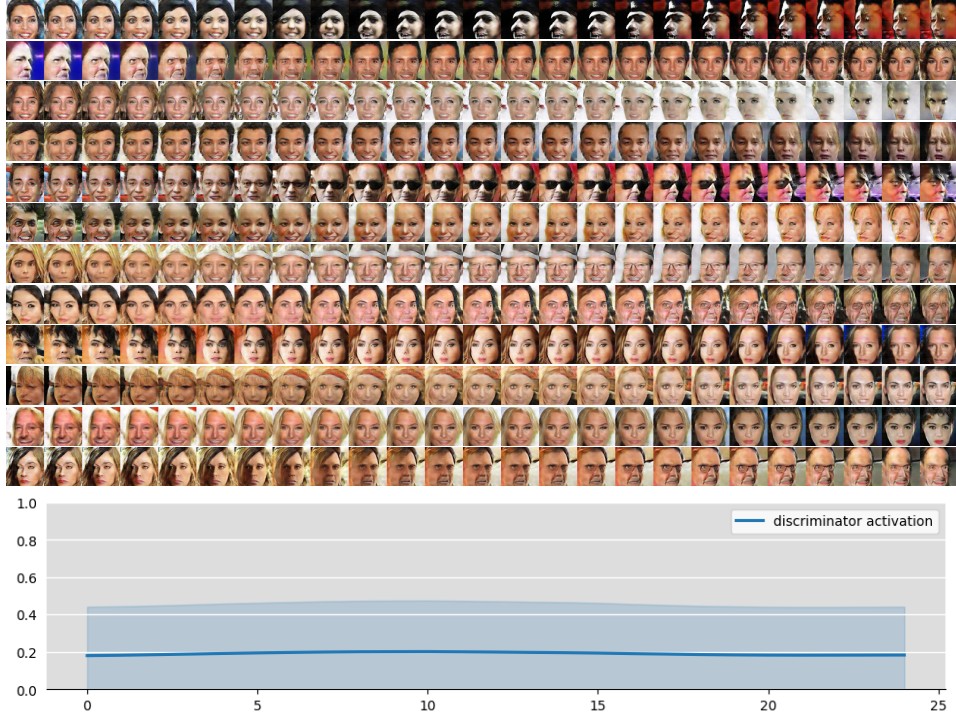

Figure 2: Sphere geodesic traversal on CelebA with a multivariate normal prior.

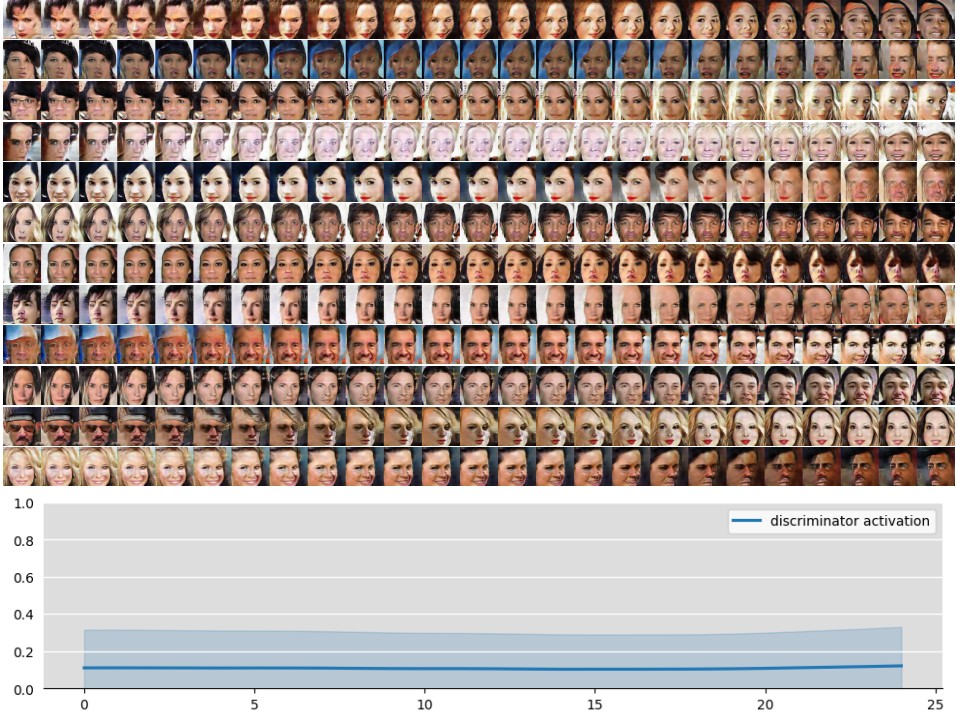

Figure 3: Sphere geodesic traversal on CelebA with a gamma prior.

### 3.2.2 STRAIGHT EUCLIDEAN TRAVERSAL

Figures 4 and 5 show traversals and discriminator activation along a straight line between the two endpoints. For the normal prior, this results in garbage, since we pass through latent space that the GAN has never seen during training. Another indication of this deficiency is the fact that the discriminator activation goes down drastically around the mid-point of the traversal.

For the gamma prior, however, the straight traversal results in a smooth interpolation between the endpoints. Note that these traversals are much more semantic in nature, with the samples along the path really lying *in between* the endpoints. Also note the emergence of a *mean sample* when looking at the mid-points. In the case of faces, these mid-points, which are points closest to the origin, tend to be very common looking faces, looking straight ahead and having little uncommon features. This emergence is even more pronounced in the traversals on the LSUN datasets in the Appendix.

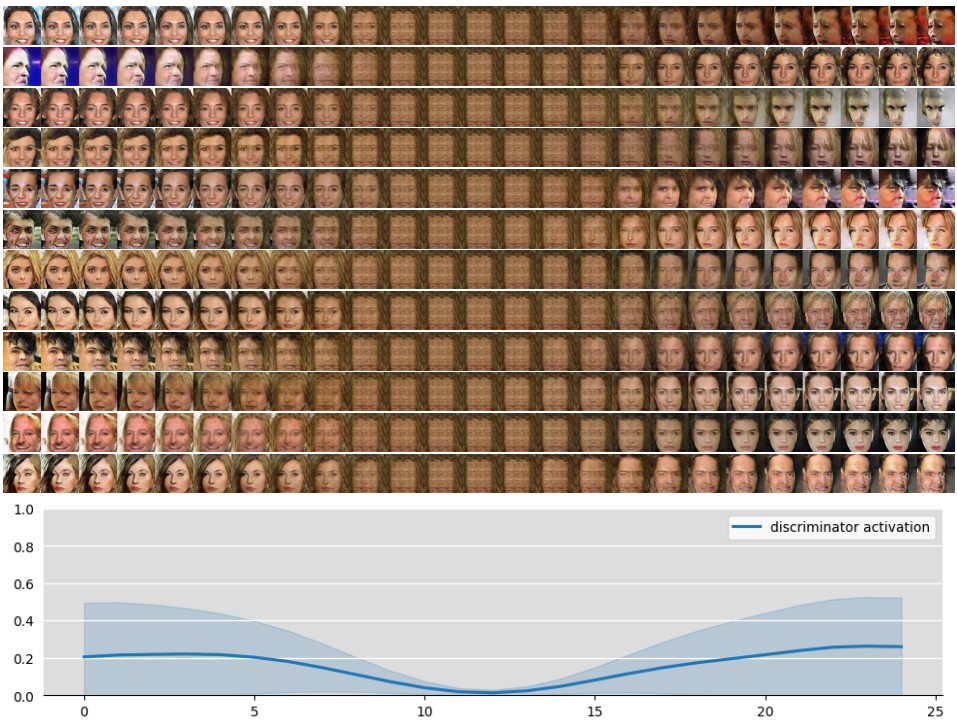

Figure 4: Straight Euclidean traversal on CelebA with a multivariate normal prior.

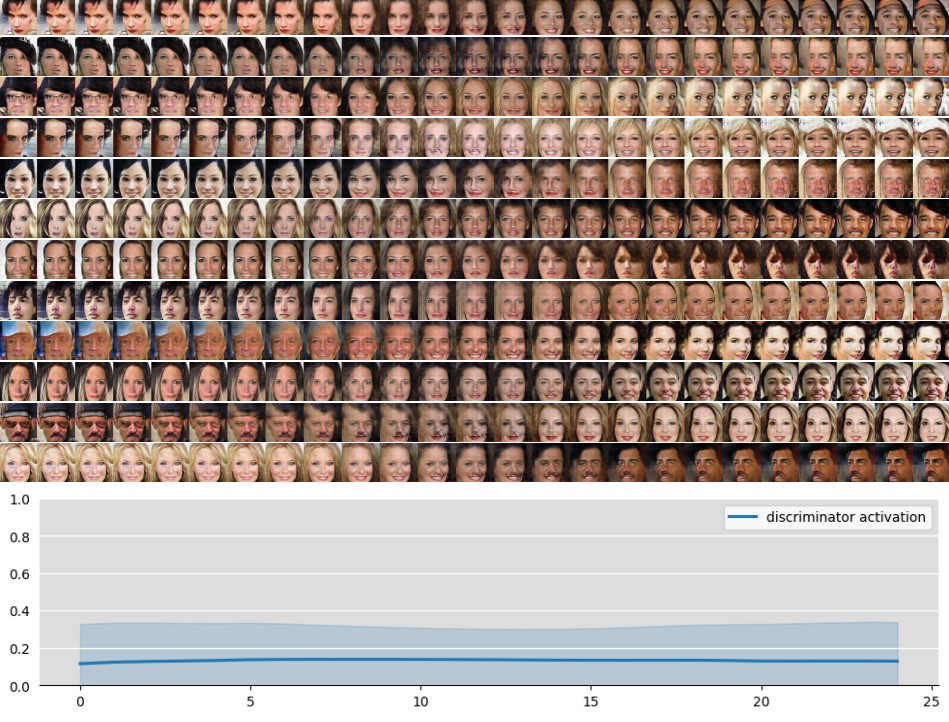

Figure 5: Straight Euclidean traversal on CelebA with a gamma prior.

### 3.3 LATENT MEAN SAMPLES

Since we noticed in our traversal experiments with gamma priors the interesting phenomenon that the mid-points of the sampled pairs of endpoints tend to converge to what one might call *mean* samples, we took our trained models and specifically sampled points close to the coordinate origin in order to directly obtain such mean samples. Figure 6 shows these mean samples for our different datasets.

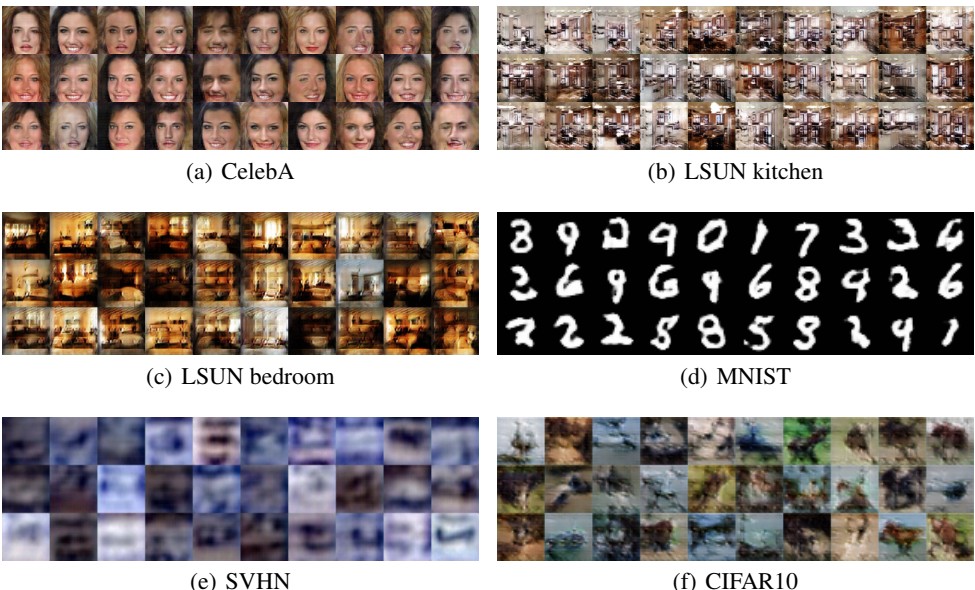

(a) CelebA

(b) LSUN kitchen

(c) LSUN bedroom

(d) MNIST

(e) SVHN

(f) CIFAR10

Figure 6: Samples around the coordinate origin in latent space on GANs trained with a gamma prior.

## 3.4 Effects of the Latent Space Dimensionality

We here empirically test the validity of the theoretical predictions made about the effect of the latent space dimensions on latent space traversals. We trained GANs for multiple latent space dimensionalities and performed straight line latent traversals using the obtained generators. Figure 7 shows the results achieved on the CelebA dataset where we observe that in low dimensions, generators trained with both the Normal and Gamma priors yield satisfying results. However, as one increases the dimensionality, generators trained using a Normal prior degrade quickly, while those trained using a Gamma prior remain unaffected. These results are therefore in accordance with the theoretical predictions made earlier. Further results on different datasets are contained in the Appendix.

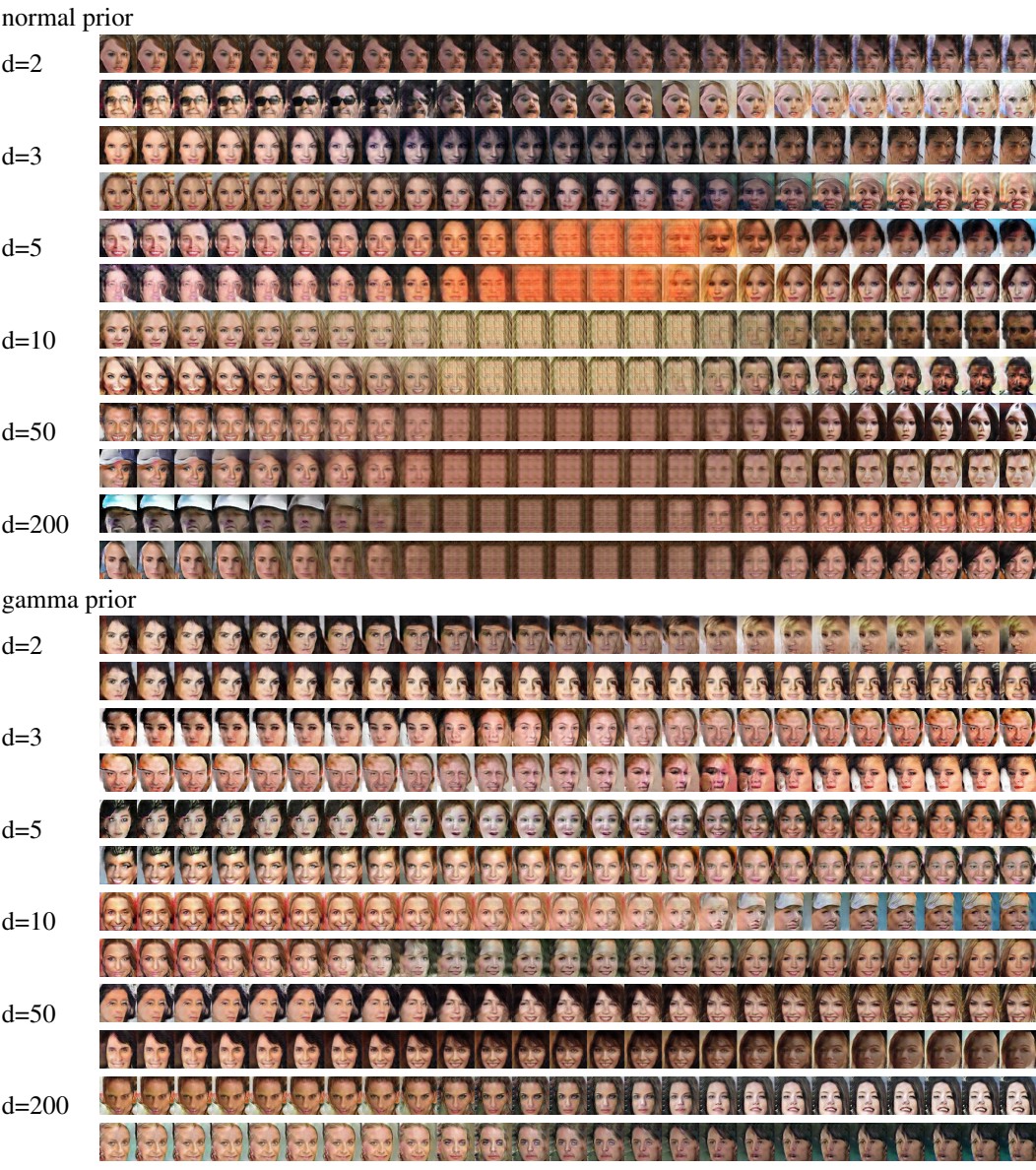

Figure 7: Straight traversal in generators with different latent space dimensionality on CelebA with a normal (top) and gamma (bottom) prior.

### 3.5 ALGEBRA EXPERIMENTS

Since we've established that using our gamma priors results in the latent space becoming more Euclidean in nature, we can ask whether this also helps for another task people are often using to evaluate generative models.

We perform various analogy experiments such as the one described in Mikolov et al. (2013b) who demonstrated words vectors exhibit relationships such as "Paris - France + Italy = Rome". In order to perform such experiments, we use the CelebA dataset that provides multiple binary attribute labels for each sample. Consider two attributes, $\mathcal{A}$ and $\mathcal{B}$. We denote by $[A, B]$ a pair of samples that have both attributes, by $[a, b]$ samples that have none of the two, and $[A, b], [a, B]$ samples that have only one of the attributes.

For each pair of attributes $\mathcal{A}, \mathcal{B}$, we want that

$$z([A, B]) - z([a, B]) + z([a, b]) \stackrel{\scriptscriptstyle\sim}{=} z([A, b]) \tag{11}$$

where $z([\bullet, \bullet])$ denotes the mean latent representation of a set of samples with (or without) the given attributes.

Using a pre-trained model, we sample a batch of samples from the generator from each of the four categories using a classifier to decide which category a sample belongs to. We then quantify to what degree the analogy described in Equation 11 holds using the following *Latent Algebra Score (LAS)*:

$$\text{LAS} = \frac{2}{N(N-1)} \sum_{\mathcal{A} \neq \mathcal{B}} \frac{\|z([A, B]) - z([a, B]) + z([a, b]) - z([A, b])\|_2^2}{m_{\|z\|_2^2}} \tag{12}$$

where $N$ is the number of binary attributes and $m_{\|z\|_2^2}$ is the mean squared norm of all used latent vectors. The results shown in Table 1 reveal that the gamma sampling procedure produces better analogies compared to the standard sampling with a normal prior.

| Prior | LAS |
|--------|----------|
| normal | 0.007496 |
| gamma | 0.005638 |

Table 1: Latent Algebra Score for CelebA (lower is better)

## 4 RELATED WORK

Learned latent representations often allow for vector space arithmetic to translate to semantic operations in the data space Radford et al. (2015); Larsen et al. (2015). Early observations showing that the latent space of a GAN finds semantic directions in the data space (e.g. corresponding to eyeglasses and smiles) were made in Radford et al. (2015). Recent work has also focused on learning better similarity metrics Larsen et al. (2015) or providing a finer semantic decomposition of the latent space Donahue et al. (2017). As a consequence, the evaluation of current GAN models is often done by sampling pair of points and linear interpolating between them in the latent space, or performing other types of noise vector arithmetic Bojanowski et al. (2017). This results in sampling the latent space from locations with very low probability mass. This observation was also made in White (2016) who suggested replacing linear interpolation with spherical linear interpolation which prevents diverging from the model's prior distribution.

## 5 CONCLUSION

While the standard way of sampling latent vectors for GANs is based on using a Normal distribution over the latent space, we showed that it might produce samples that are not likely under the model distribution. We discussed how this procedure suffers from the curse of dimensionality and demonstrated how a simple alternative procedure solves this problem. Finally, we provided an extensive set of experiments that clearly demonstrate visual improvements in the samples generated using our gamma sampling procedure.

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

# A  MORE TRAVERSAL EXPERIMENTS

## A.1  STRAIGHT EUCLIDEAN TRAVERSAL

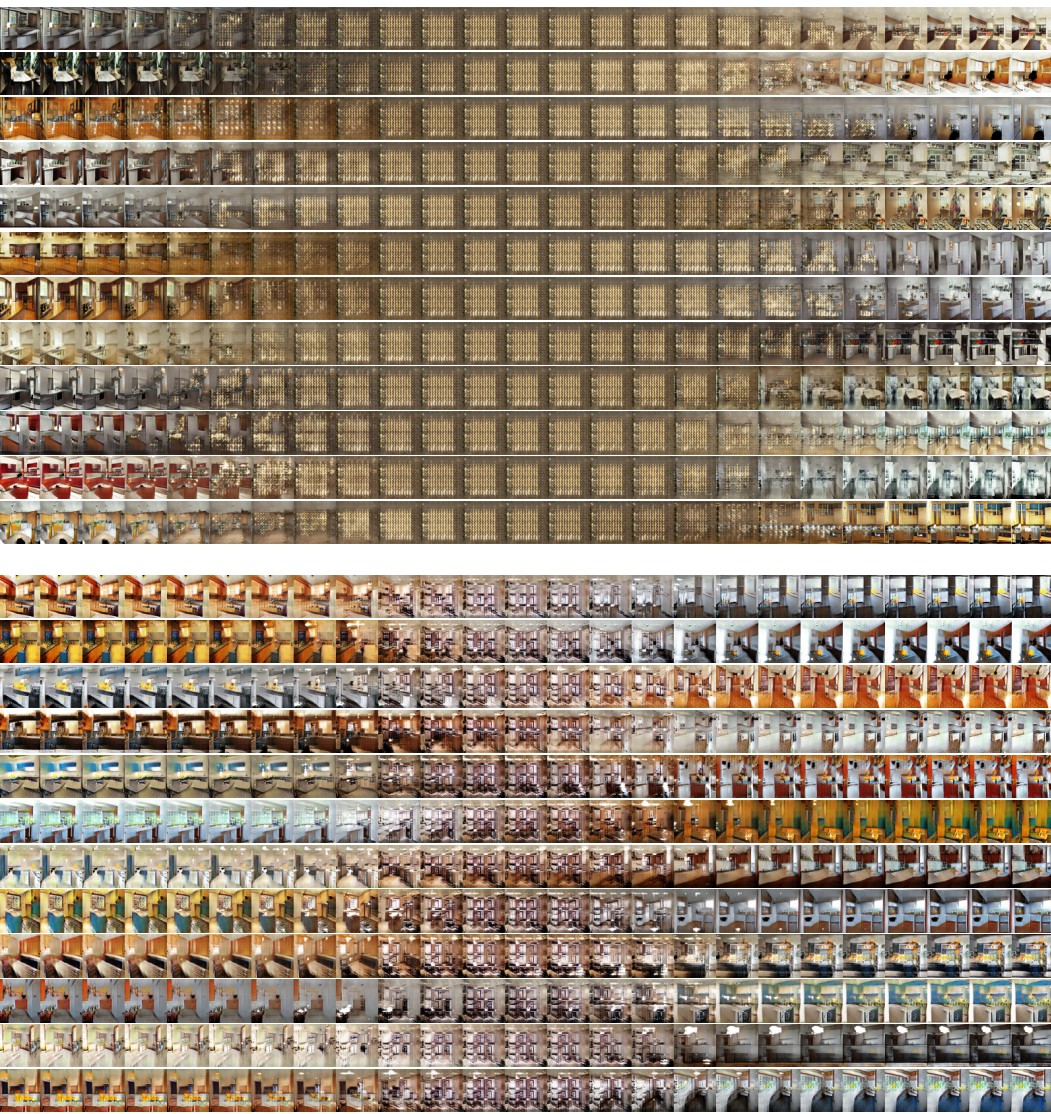

Figure 8: Straight Euclidean traversal on LSUN kitchen with a normal (top) and gamma (bottom) prior.

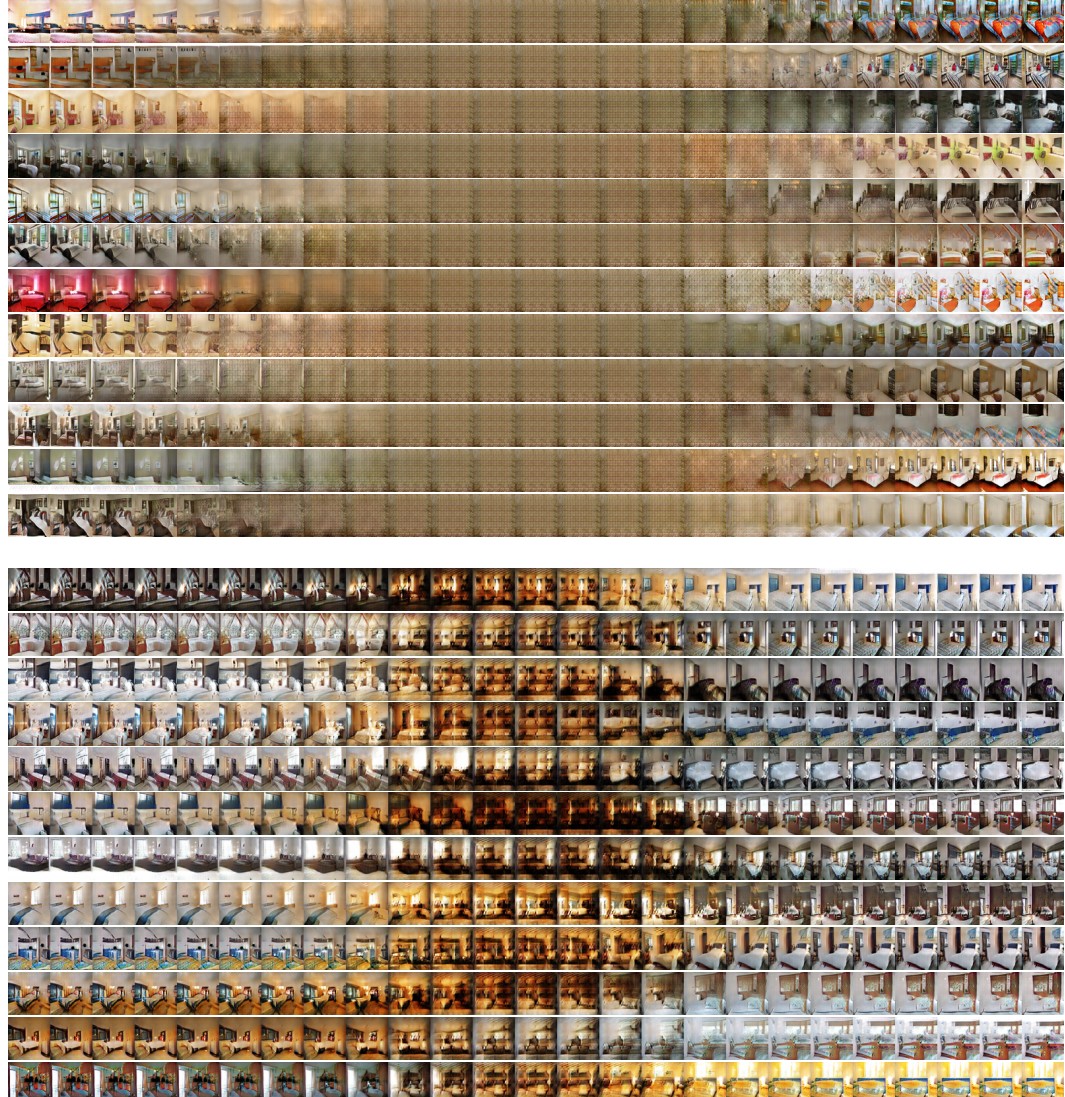

Figure 9: Straight Euclidean traversal on LSUN bedroom with a normal (top) and gamma (bottom) prior.

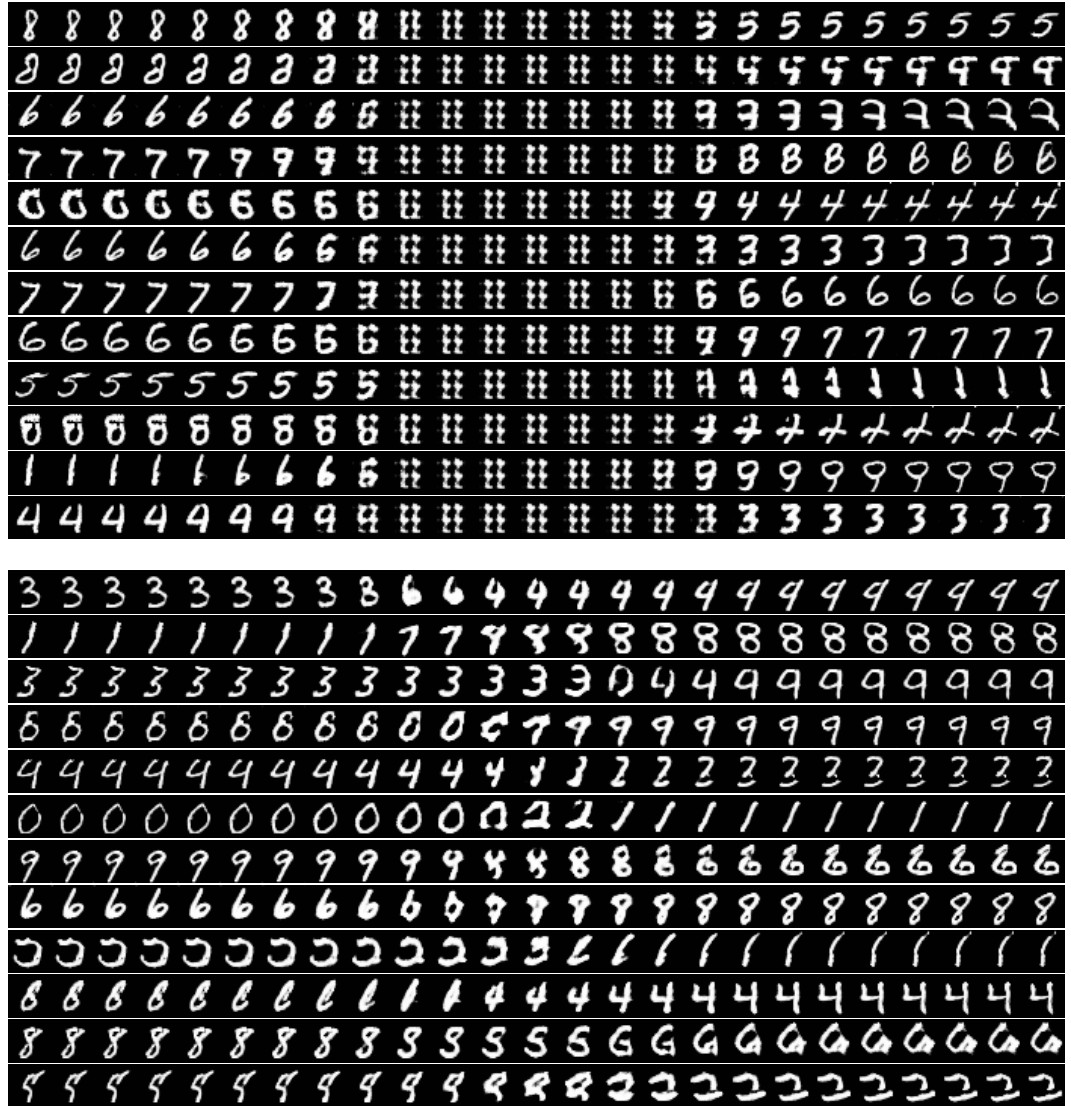

Figure 10: Straight Euclidean traversal on MNIST with a normal (top) and gamma (bottom) prior.

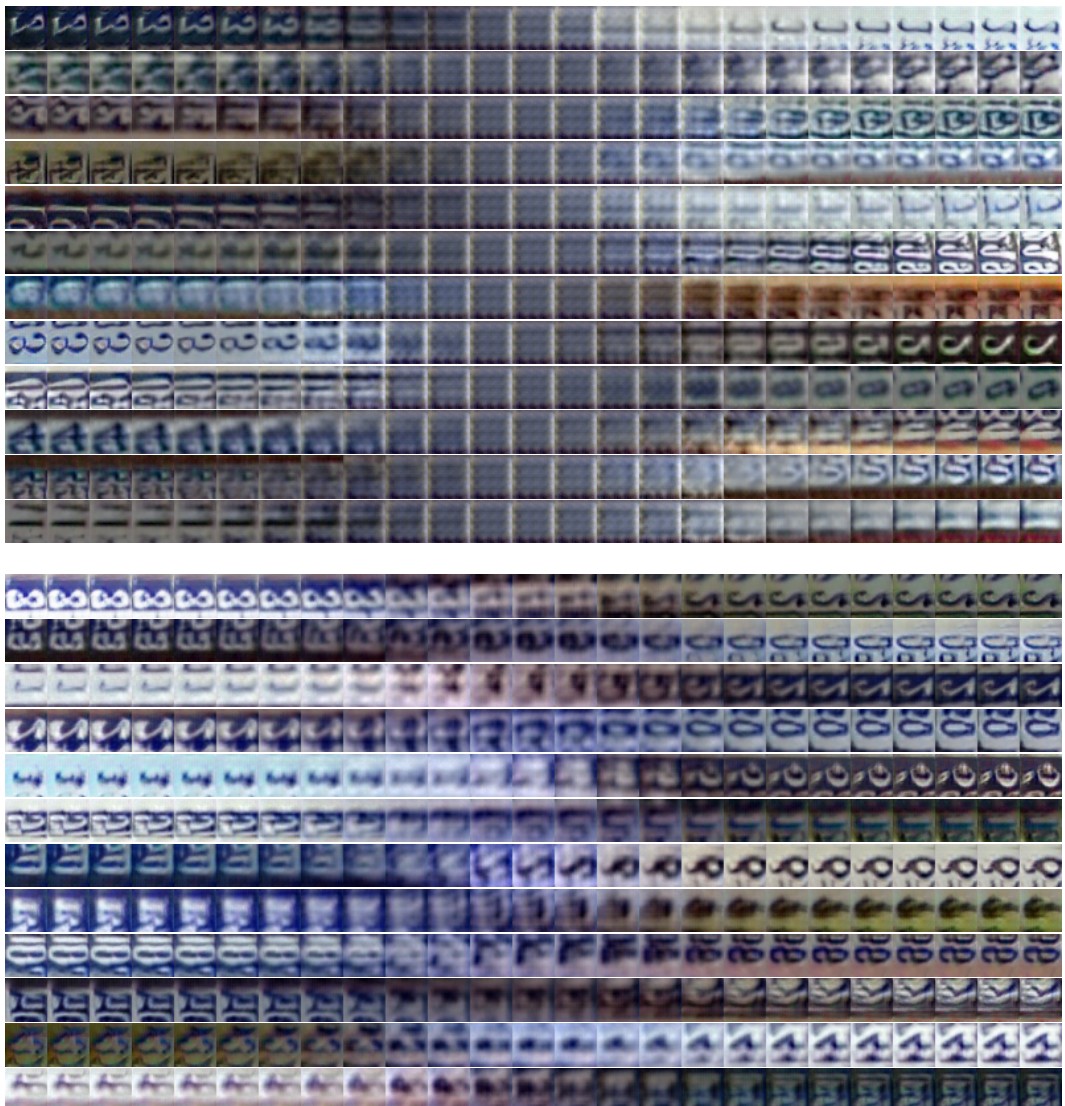

Figure 11: Straight Euclidean traversal on SVHN with a normal (top) and gamma (bottom) prior.

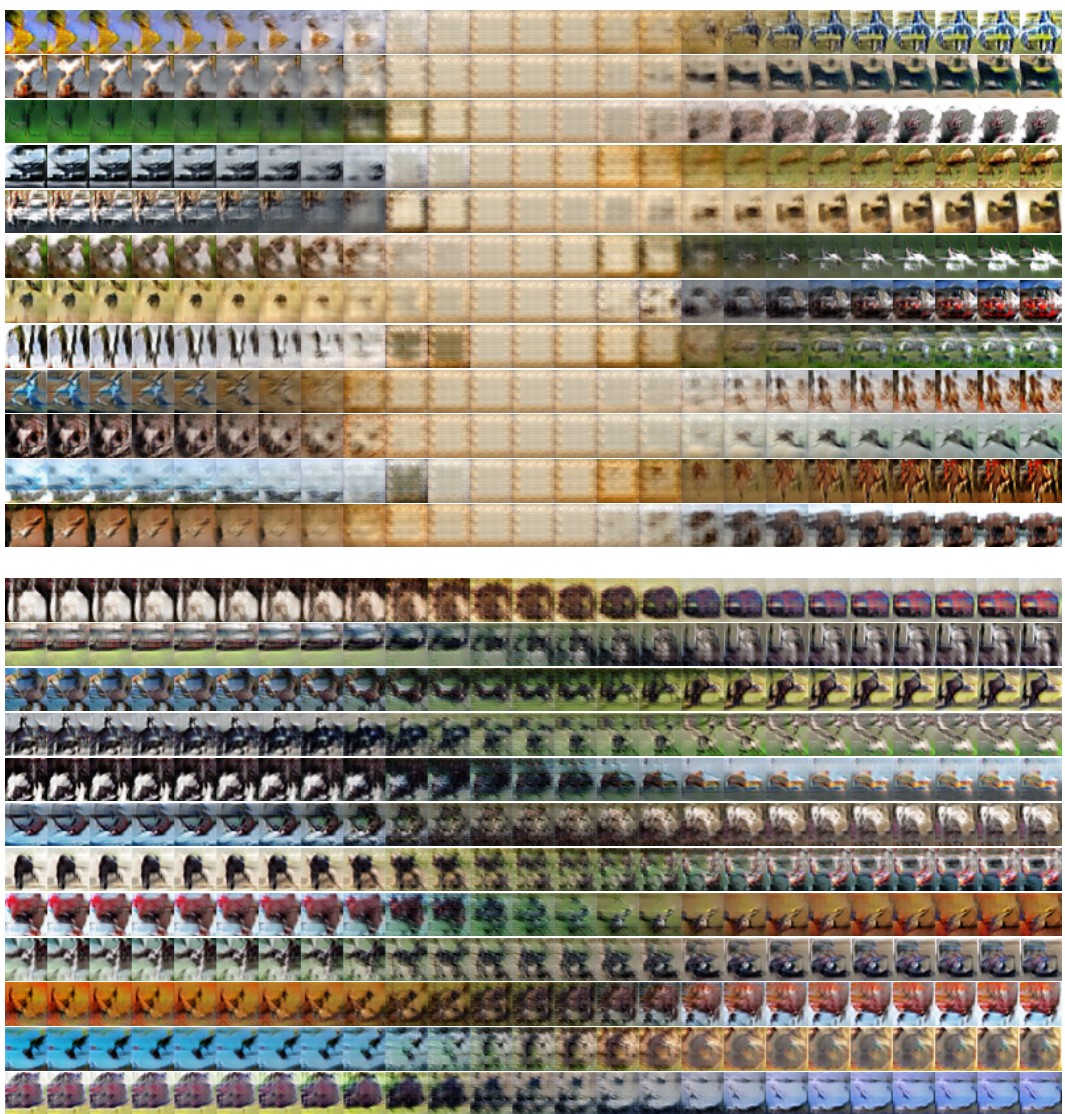

Figure 12: Straight Euclidean traversal on CIFAR10 with a normal (top) and gamma (bottom) prior.

## A.2 SPHERE GEODESIC TRAVERSAL

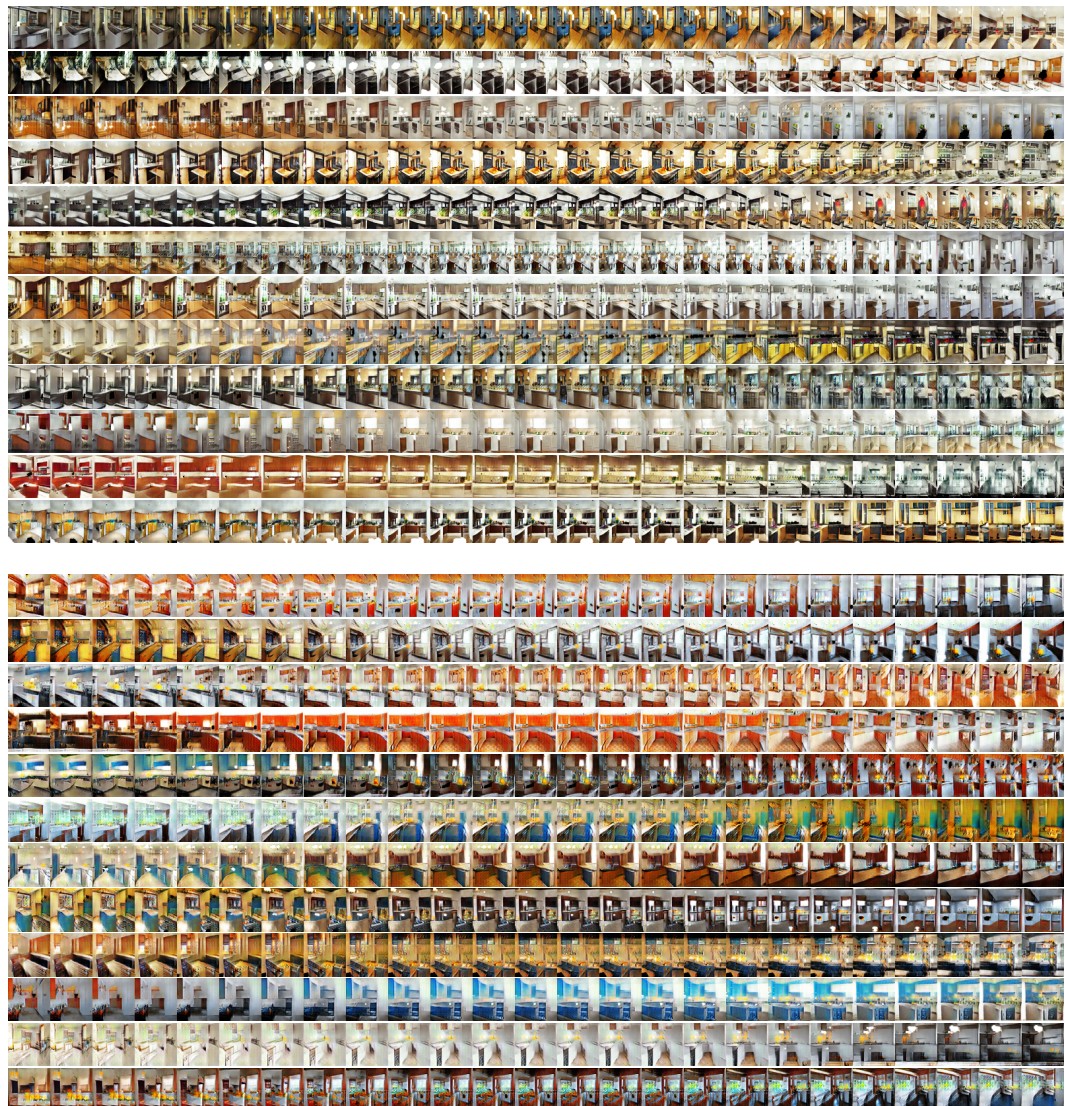

Figure 13: Sphere geodesic traversal on LSUN kitchen with a normal (top) and gamma (bottom) prior.

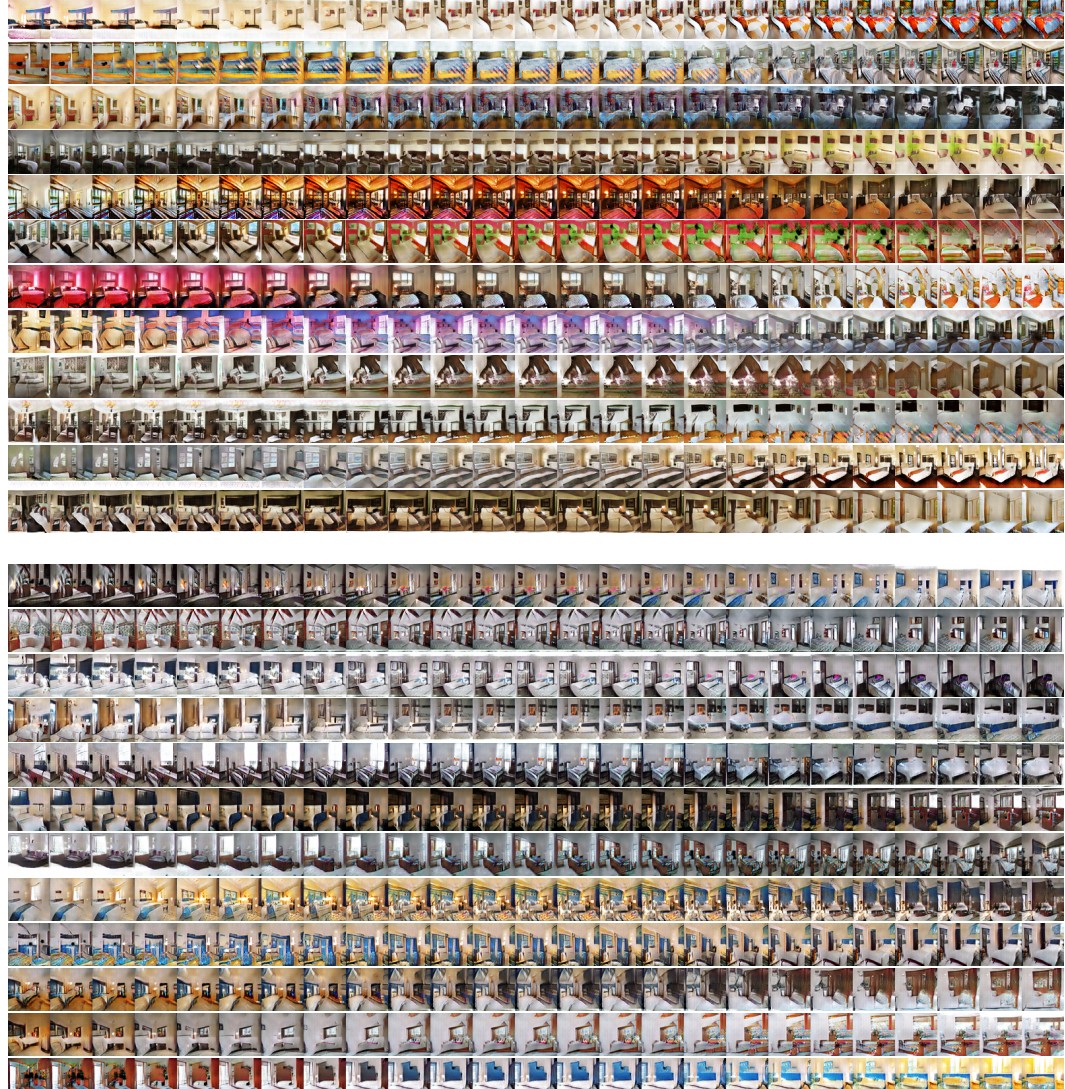

Figure 14: Sphere geodesic traversal on LSUN bedroom with a normal (top) and gamma (bottom) prior.

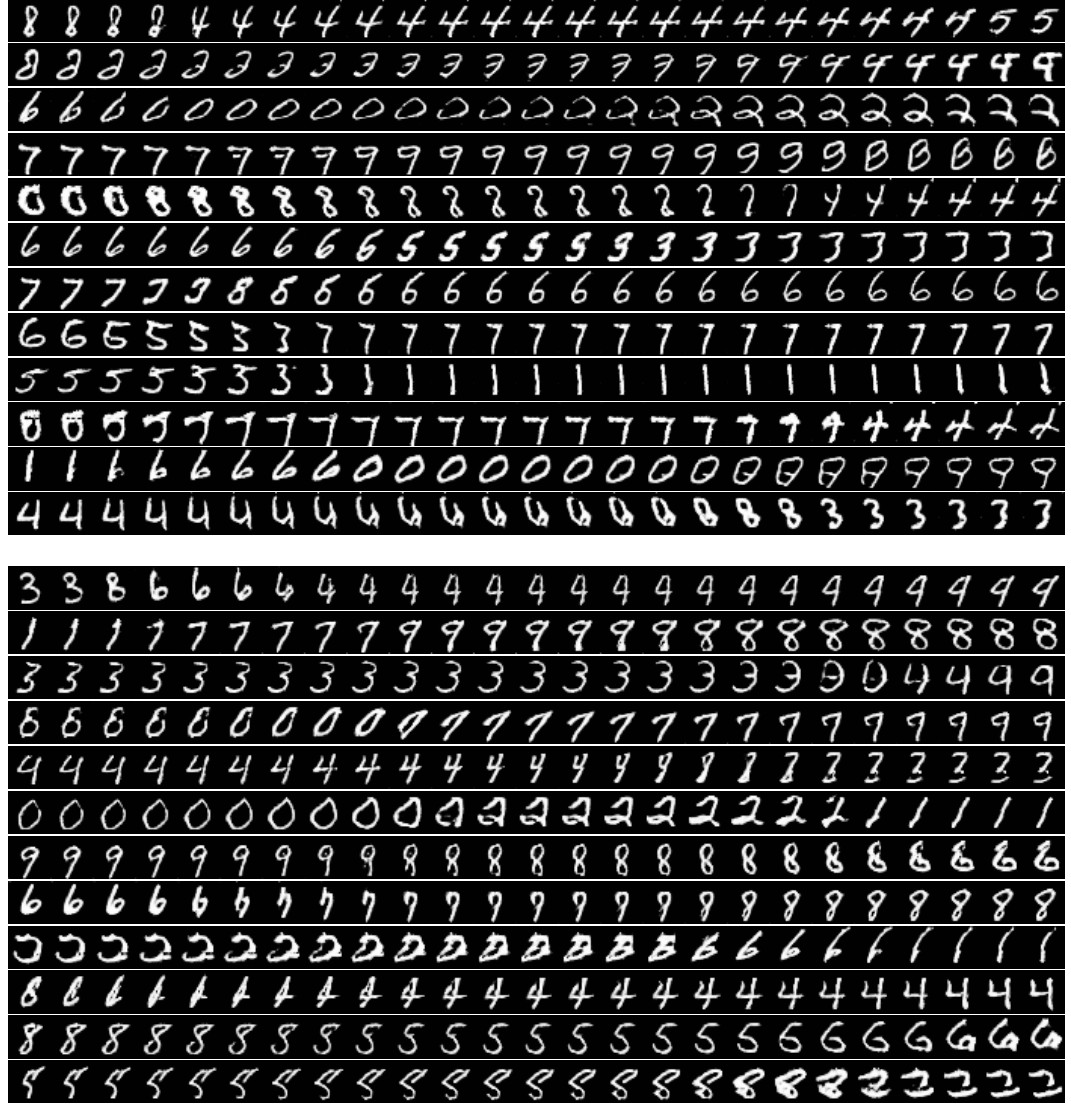

Figure 15: Sphere geodesic traversal on MNIST with a normal (top) and gamma (bottom) prior.

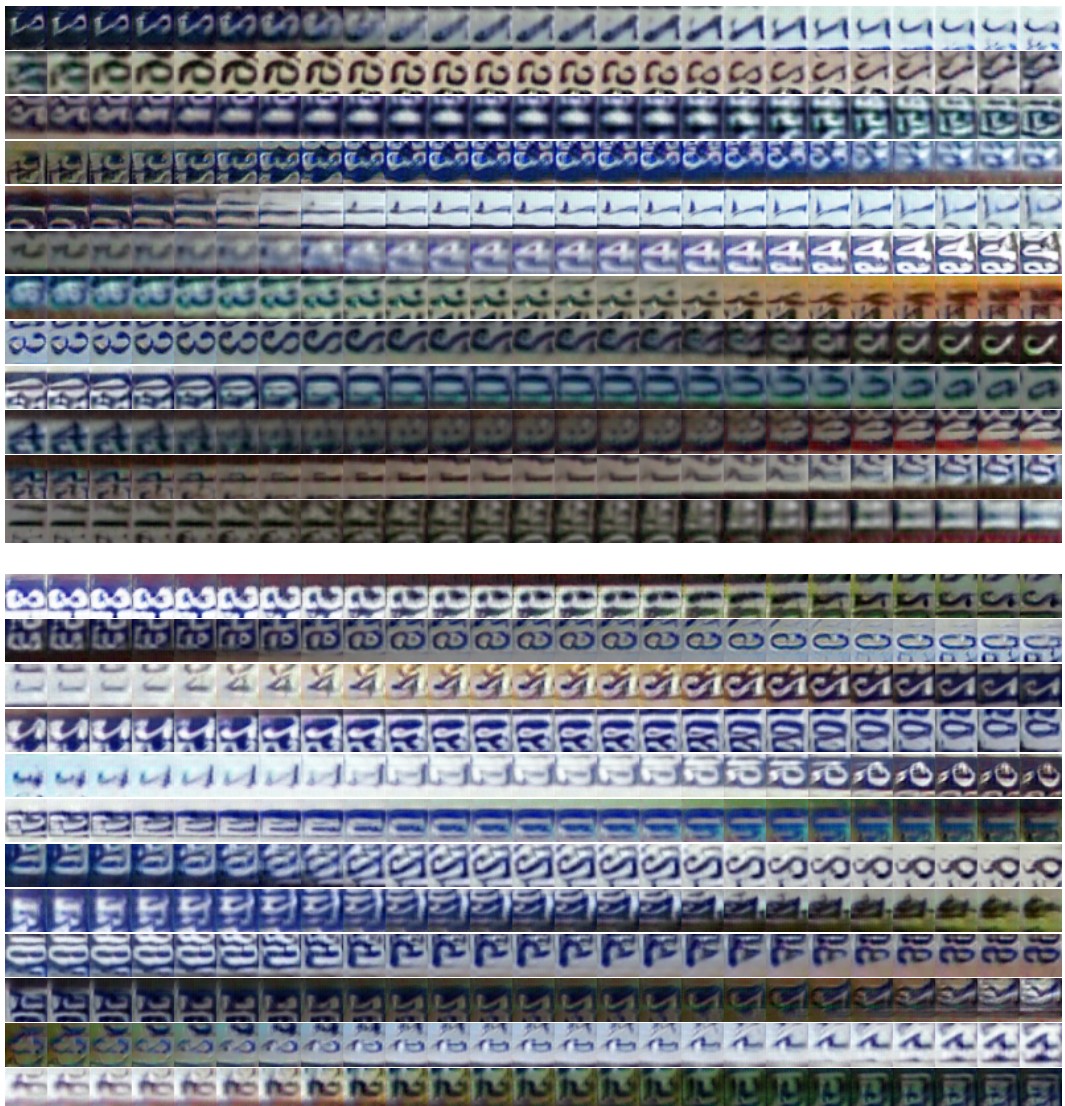

Figure 16: Sphere geodesic traversal on SVHN with a normal (top) and gamma (bottom) prior.

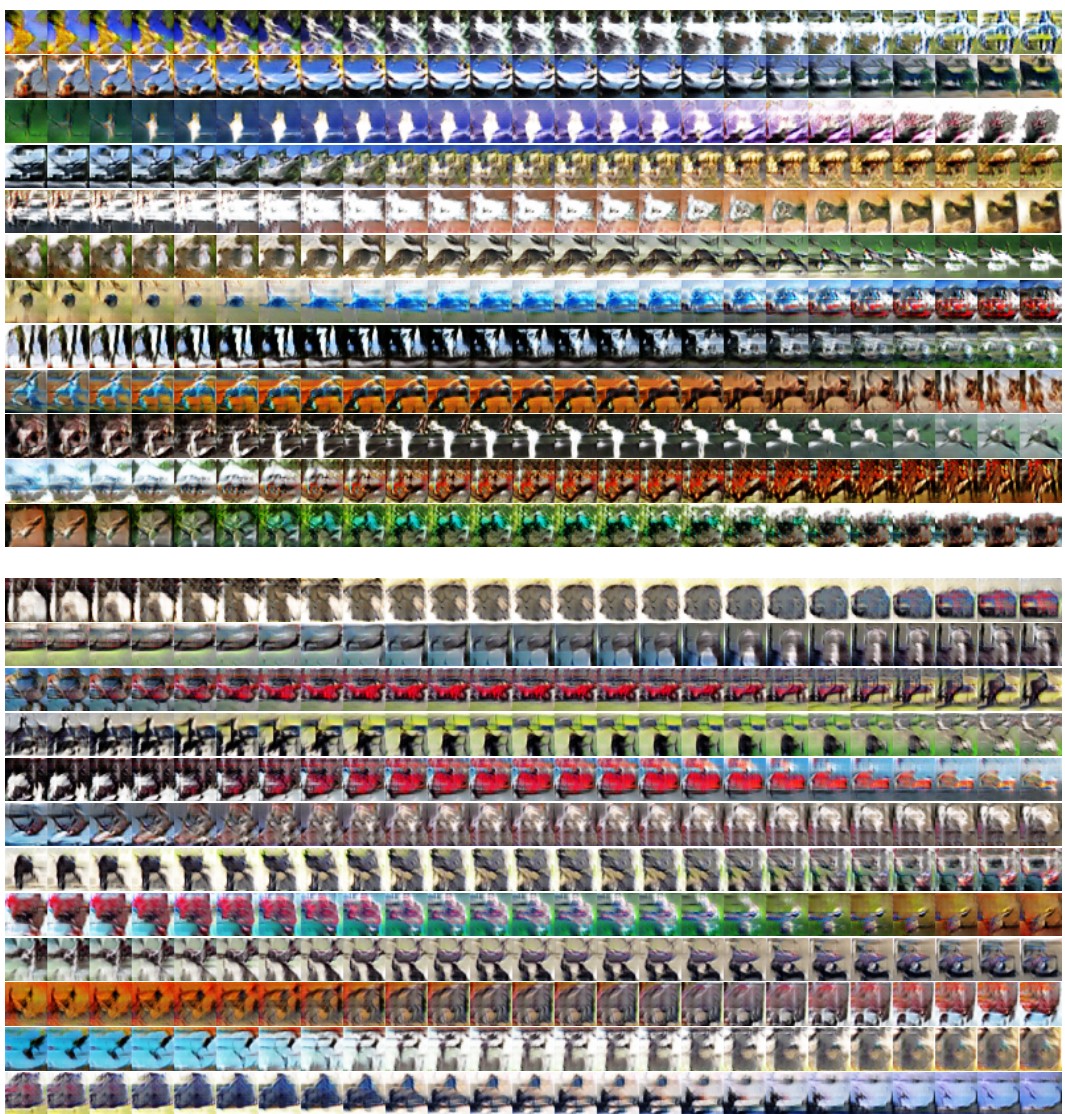

Figure 17: Sphere geodesic traversal on CIFAR10 with a normal (top) and gamma (bottom) prior.

## B  MORE EXPERIMENTS COMPARING LATENT SPACE DIMENSIONALITY

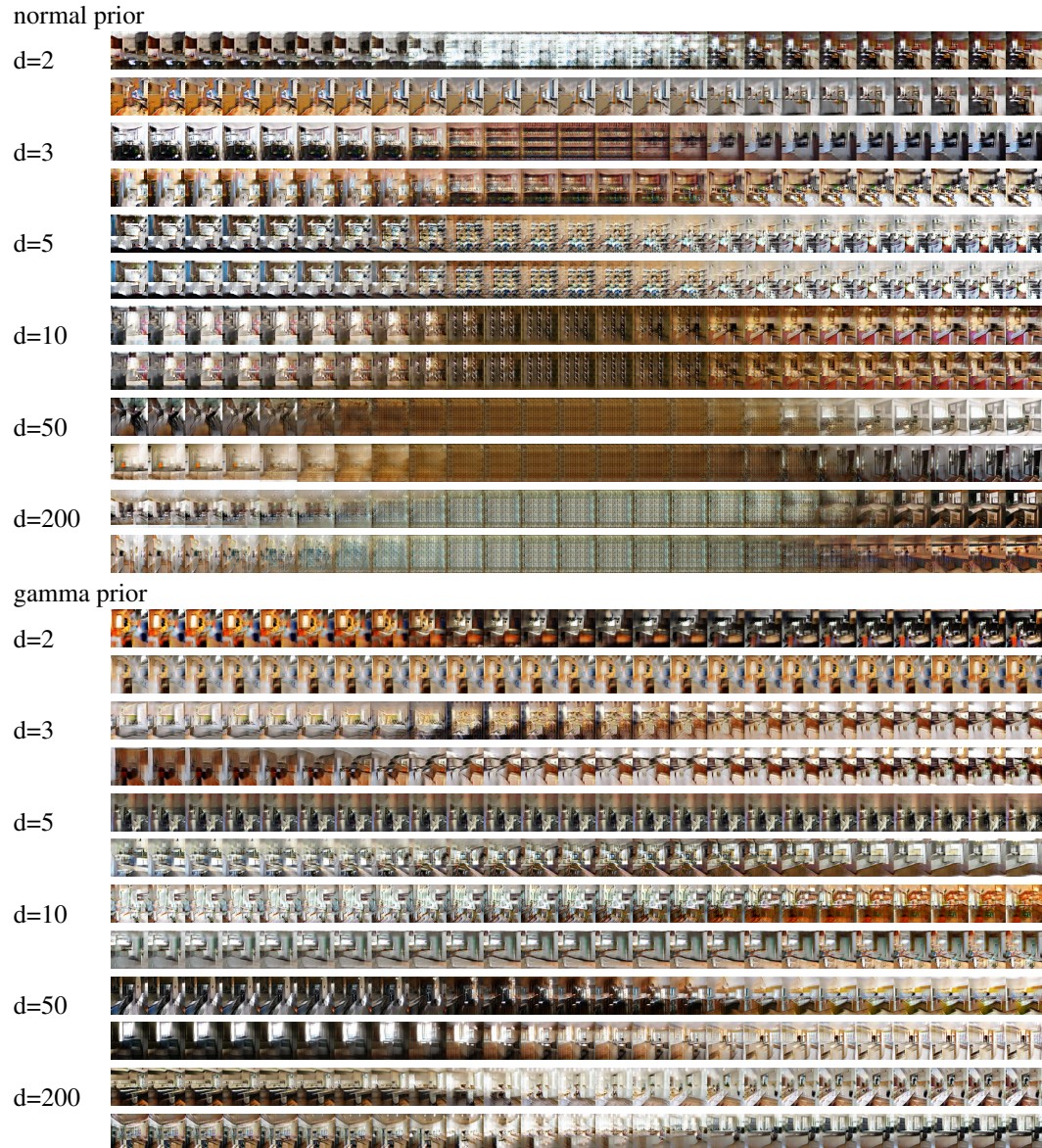

Table 2: Straight traversal in generators with different latent space dimensionality on LSUN kutchen with a normal (top) and gamma (bottom) prior.

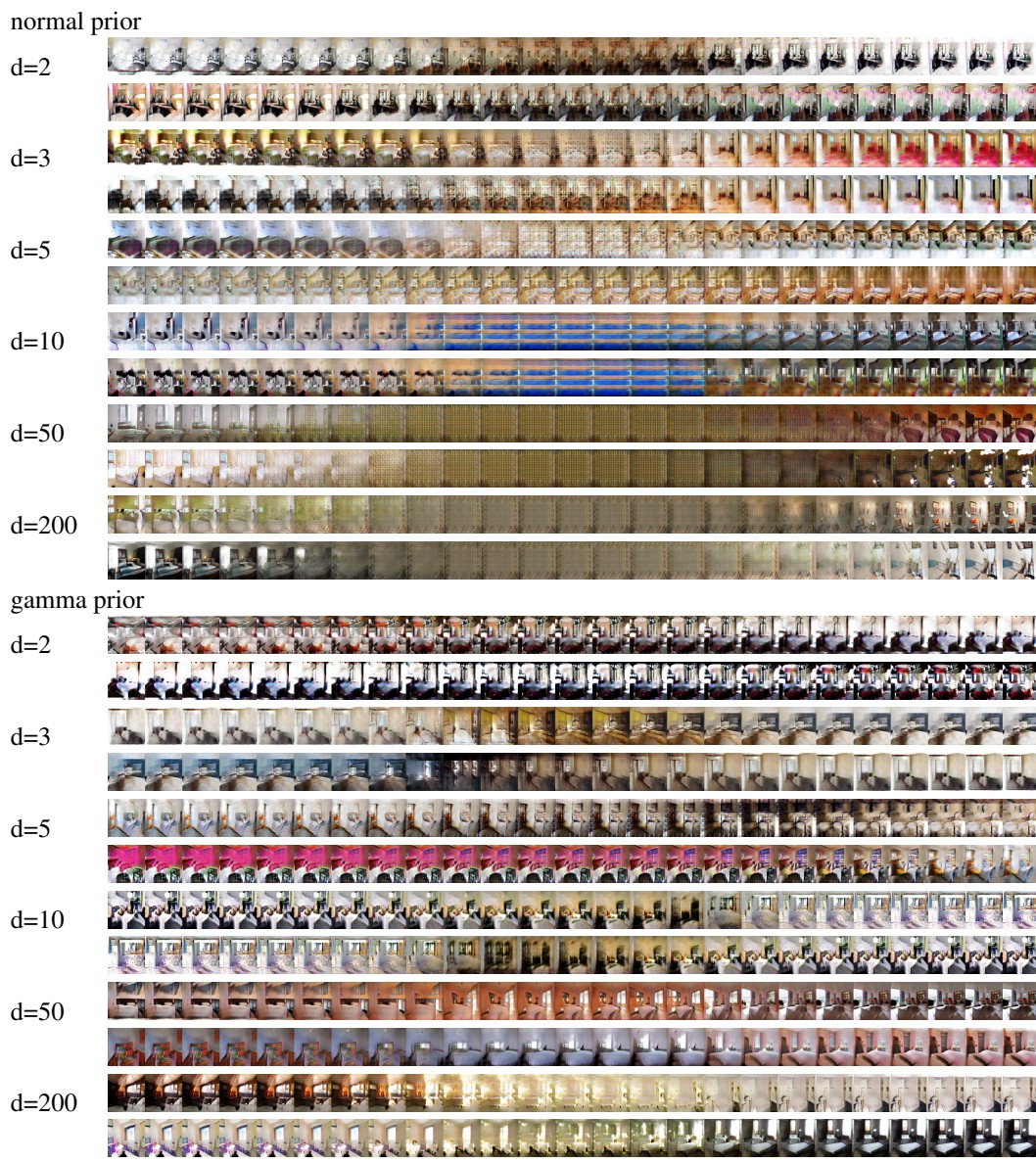

Table 3: Straight traversal in generators with different latent space dimensionality on LSUN bedroom with a normal (top) and gamma (bottom) prior.

normal prior

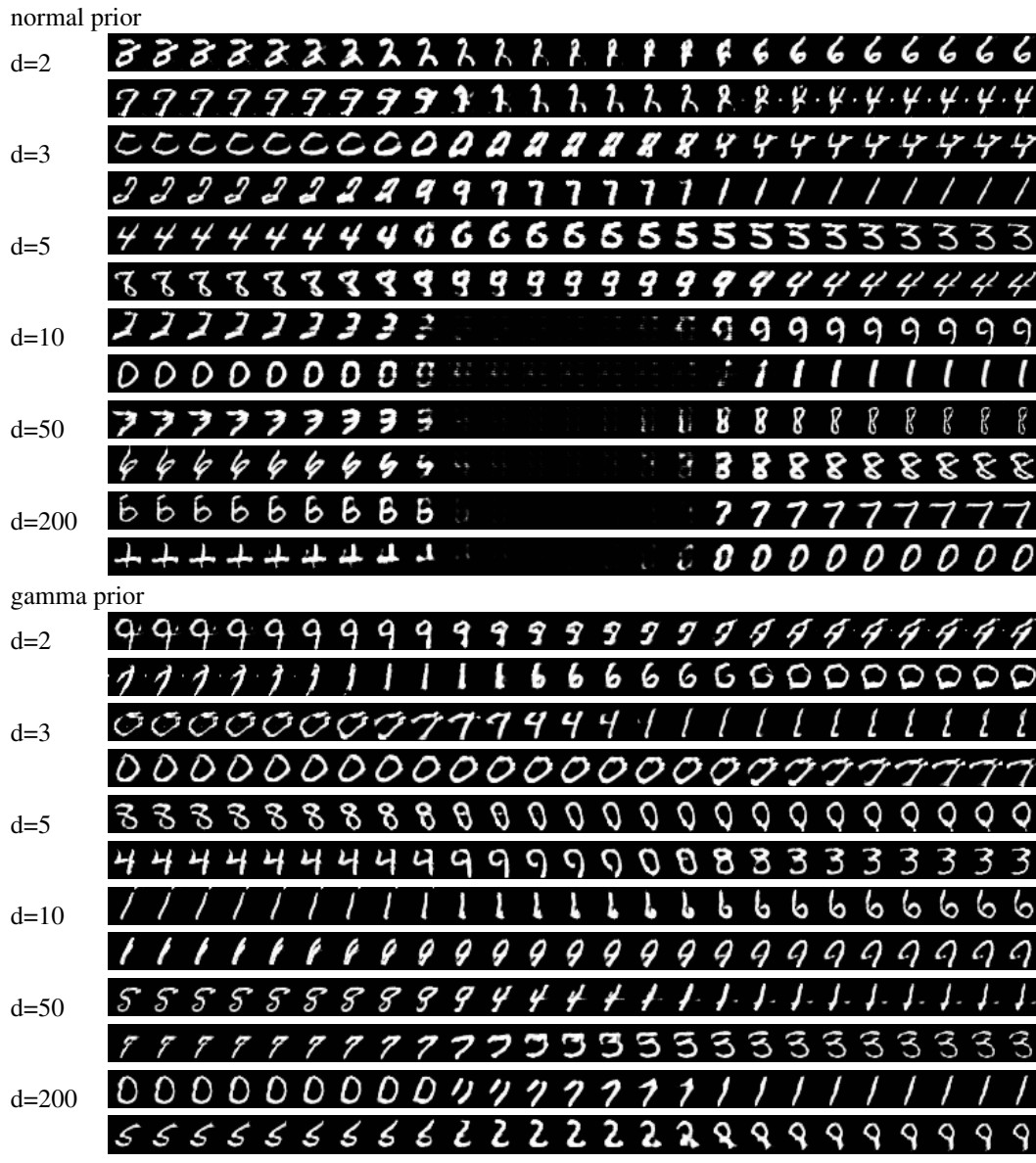

Table 4: Straight traversal in generators with different latent space dimensionality on MNIST with a normal (top) and gamma (bottom) prior.

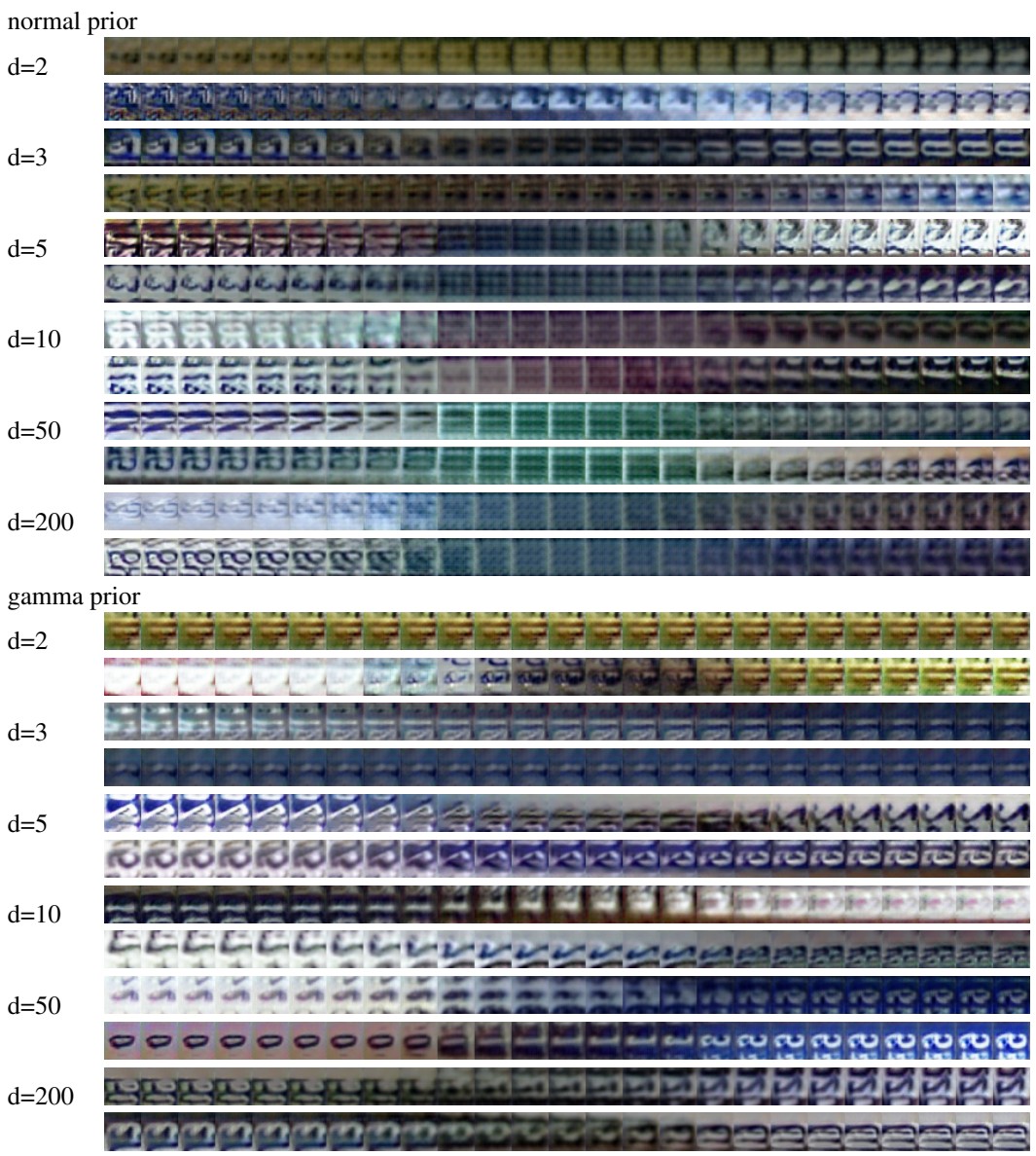

Table 5: Straight traversal in generators with different latent space dimensionality on SVHN with a normal (top) and gamma (bottom) prior.

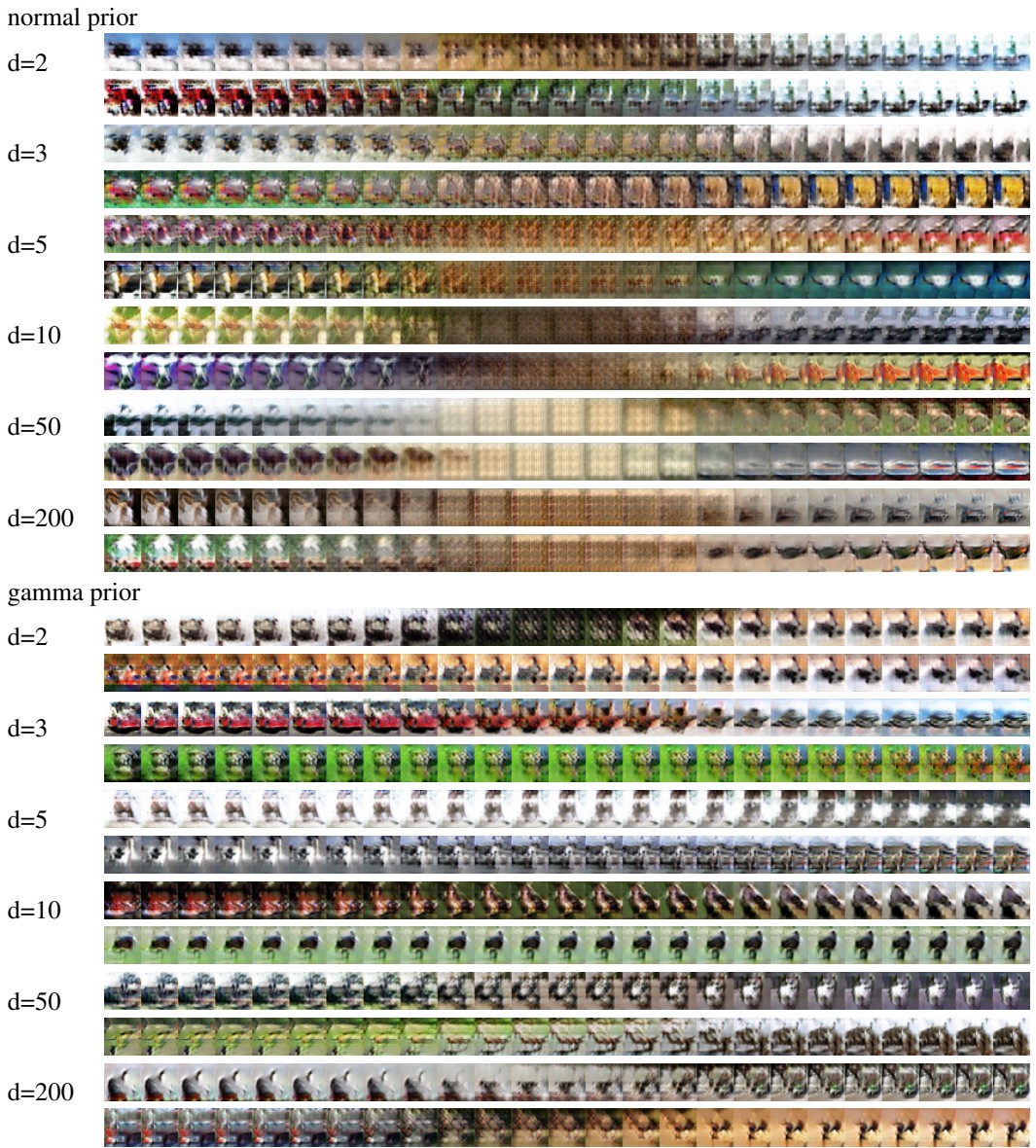

Table 6: Straight traversal in generators with different latent space dimensionality on CIFAR10 with a normal (top) and gamma (bottom) prior.

## C  EXPERIMENT SETUP

For our experiments, we use a standard DCGAN architecture featuring 5 deconvolutional and convolutional layers with $4\times4$ filters applied in strides of 2 in the generator and discriminator, respectively. We use ReLU nonlinearities and batch normalization in the generator, while the discriminator features leaky ReLU nonlinearities and batch normalization from the 2nd layer on. The latent space for all models is of dimension 100 and the scale parameters for both the normal and gamma distributions are set to $1.0$. The networks are trained using RMSProp with a learning rate of $0.0003$ and mini-batches of size 100.

The samples for the CelebA dataset have been cropped to $118 \times 118$ and then resized to $64 \times 64$.

