# OpenReview forum: "Semantic Interpolation in Implicit Models"
_ICLR.cc/2018/Conference — Accept (Poster)_

### Official Review · AnonReviewer3 · 2017-11-27
**code distributions for implicit models**

**Rating:** 6
**Confidence:** 3

**Review:**

The paper concerns distributions used for the code space in implicit models, e.g. VAEs and GANs. The authors analyze the relation between the latent space dimension and the normal distribution which is commonly used for the latent distribution. The well-known fact that probability mass concentrates in a shell of hyperspheres as the dimensionality grows is used to argue for the normal distribution being sub-optimal when interpolating between points in the latent space with straight lines. To correct this, the authors propose to use a Gamma-distribution for the norm of the latent space (and uniform angle distribution). This results in more mass closer to the origin, and the authors show both that the midpoint distribution is natural in terms of the KL divergence to the data points, and experimentally that the method gives visually appealing interpolations.

While the contribution of using a standard family of distributions in a standard implicit model setup is limited, the paper does make interesting observations, analyses and an attempt to correct the interpolation issue. The paper is clearly written and presents the theory and experimental results nicely. I find that the paper can be accepted but the incremental nature of the contribution prevents a higher score.

---

> ### Author Response · Authors · 2018-01-02
> **Thank You**
>
> Dear Reviewer,
> Thank you for your positive review.

---

### Official Review · AnonReviewer2 · 2017-11-27
**Neat idea, needs more work.**

**Rating:** 5
**Confidence:** 4

**Review:**

The authors discuss a direct Gamma sampling method for the interpolated samples in GANs, and show the improvements over usual normal sampling for CelebA, MNIST, CIFAR and SVHN datasets.

The method involves a nice, albeit minor, trick, where the chi-squared distribution of the sum of the z_{i}^{2} has its dependence on the dimensionality removed. However I am not convinced by the distribution of \|z^\prime\|^{2} in the first place (eqn (2)): the samples from the gaussian will be approximately orthogonal in high dimensions, but the inner product will be at least O(1). Thus although the \|z_{0}\|^{2} and \|z_{1}\|^{2} are chi-squared/gamma, I don't think \|z^\prime\|^{2} is exactly gamma in general.

The experiments do show that the interpolated samples are qualitatively better, but a thorough empirical analysis for different dimensionalities would be welcome. Figures 2 and 3 do not add anything to the story, since 2 is just a plot of gamma pdfs and 3 shows the difference between the constant KL and the normal case that is linear in d.

Overall I think the trick needs to be motivated better, and the experiments improved to really show the import of the d-independence of the KL. Thus I think this paper is below the acceptance threshold.

---

> ### Author Response · Authors · 2018-01-02
> **Response to feedback**
>
> We thank the reviewer for their feedback and answer their concerns and requests below.
>
> 1. distribution of \| z^\prime \|
> Thank you for pointing this out, we made a slight change to our original submission to clarify this point and corrected a minor mistake concerning the degrees of freedom of the Gamma distribution (which incorrectly had an additional factor of 2). Nevertheless, we would like to emphasize that, as claimed in our original submission, the distribution of \| z^\prime \| is indeed a gamma distribution. This can be seen as follows. First recall that for any z_0 and z_1 vectors drawn from a Gaussian distribution, their squared lengths follows a gamma distribution (this is just the definition of a Gamma distribution which models a sum of the squares of independent standard normal random variables). Then consider the average (z_0 + z_1) / 2 discussed in the paper, this average is then again a gaussian vector (since the sum of two independent normally distributed random variables is normal), so its squared length must also be gamma. Note that the same applies if we scale z_i with a factor sqrt(gamma)/||z_i||. We’ve adjusted the method section for these corrections.
>
> 2. “Thorough empirical analysis for different dimensionalities would be welcome”
> We have now added a new section named “Effects of the Latent Space Dimensionality” in the experiments section (we also provide more results in the appendix) where we show examples of straight traversals for GANs trained using different latent dimensionalities. We observe that for low dimensional latent spaces, both the normal and gamma priors produce results where the interior regions seem to produce meaningful samples. However, as the dimensionality grows, the mid-points in the normal-prior GANs quickly degrade, whereas the GANs trained using the gamma prior do not.
>
> 3. “Figures 2 and 3 do not add anything to the story”
> We agree, they merely served to illustrate points that were already made. We have removed the figures.
>
> 4. “Trick needs to be motivated better and the experiments improved to really show the improvement of the d-independence of the KL”
> See answer above, we think the newly added section in the experiments does show a clear improvement in terms of independence to the latent dimension. These results are also in accordance with the theoretical predictions made in the paper and we, therefore, believe that both the theory and experiments do motivate the use of the Gamma prior we advocate in the paper.

---

### Official Review · AnonReviewer1 · 2017-11-29
**In general, the proposed work is very interesting and the idea is neat. It is a useful contribution to the community of GANs and implicit generative models. I am impressed with the structure and presentation of the paper. Easy to follow and well supported.**

**Rating:** 7
**Confidence:** 4

**Review:**

The authors propose the use of a gamma prior as the distribution over
the latent representation space in GANs. The motivation behind it is that
in GANs interpolating between sampled points is common in the process of generating examples but the use of a normal prior results in samples that fall in low probability mass regions. The use of the proposed gamma distribution, as a simple alternative, overcomes this problem.

In general, the proposed work is very interesting and the idea is neat.
The paper is well presented and I want to underline the importance of this.
The authors did a very good job presenting the problem, motivation and solution in a coherent fashion and easy to follow.

The work itself is interesting and can provide useful alternatives for the distribution over the latent space.

---

> ### Author Response · Authors · 2018-01-02
> **Thanks**
>
> Dear Reviewer,
> Thank you for your positive review.

---

### Public Comment · ~Christian_A_Naesseth1 · 2017-11-27
**Interesting!**

This seems very interesting. A quick question, do you think it would be useful to also learn the rate parameter of your Gamma distribution rather than fixing it? This could be achieved by e.g.
Naesseth, Ruiz, Linderman, Blei, "Reparameterization Gradients through Acceptance-Rejection Sampling Algorithms", 2017.

---

> ### Author Response · Authors · 2018-01-02
> **Interesting Reference**
>
> Thank you for your observation. We do think this would be an interesting direction for extending our work.

---

### Decision · Program_Chairs · 2018-01-29
**ICLR 2018 Conference Acceptance Decision**

**Decision:**

Accept (Poster)

**Comment:**

The paper presents a modified sampling method for improving the quality of interpolated samples in deep generative models.

There is not a great amount of technical contributions in the paper, however it is written in a very clear way, makes interesting observations and analyses and shows promising results. Therefore, it should be of interest to the ICLR community.